# Empirical NOx Removal Analysis of Photocatalytic Construction Materials at Real-Scale

**DOI:** 10.3390/ma14195717

**Published:** 2021-09-30

**Authors:** Miyeon Kim, Hyunggeun Kim, Jinchul Park

**Affiliations:** 1SH Urban Research Center, Seoul Housing and Communities Corporation, Seoul 06336, Korea; mykim1983@naver.com (M.K.); hgkim@i-sh.co.kr (H.K.); 2Department of Architectural Engineering, Chung-Aug University, Seoul 06974, Korea

**Keywords:** photocatalysis, construction materials, ISO22197-1, titanium dioxide (TiO_2_), nitrogen oxides (NOx)

## Abstract

The NOx removal performance of photocatalytic construction materials is demonstrated using two experiments under indoor and outdoor environments: (1) A photoreactor test was conducted to assess the NO removal performance of construction materials (e.g., coatings, paints and shotcrete) using a modified ISO 22197-1 method; (2) A water washing test was conducted using two specimens enlarged to the size of actual building materials and artificially exposed to NOx in a laboratory to analyze NOx removal performance. For (1), the UV irradiation of the outdoor environment was analyzed and the experiment was conducted in an indoor laboratory under UV irradiation identical to that of the outdoor condition. Photoreactor tests were conducted on construction materials applied to actual buildings located in Seoul, South Korea. In (2), the enlarged specimen was used for a field experiment by applying a modified method from the ISO 22197-1 standard. On sunny days, the NOx removal performance (3.12–4.76 μmol/150 cm^2^·5 h) was twice as much as that of the ISO 22197-1 standard specification (2.03 μmol/150 cm^2^·5 h) in the real-world. The washing water test results indicated that general aqueous paint achieved a NOx removal of 3.88 μmol, whereas photocatalytic paint was superior to 14.13 μmol.

## 1. Introduction

Recently, the emission of nitrogen oxide (NOx) has been considered a major environmental concern. NOx not only causes respiratory disorders in the human body, but also destroys the ozone layer in the stratosphere, thereby promoting climate change [1,2]. Further, NOx is a precursor substance of particulate matter generated via chemical reactions with other precursors [3,4,5]. Thus, NOx abatement regulation based on various air pollution policies established by the Gothenburg and Kyoto Protocols has contributed to a decrease in the worldwide emissions of NOx [6]. In addition to regulating NOx generation from automobiles, power plants and manufacturing facilities [7], technologies that remove NOx in the atmosphere play a vital role in NOx abatement [8,9,10,11,12]. Heterogeneous photocatalysis is a promising NOx removal technology that has been applied as a construction material [13,14]. Titanium dioxide (TiO_2_) is among the most widely used photocatalytic nanoparticles for such applications [15]. The NOx removal process using TiO_2_ involves the illumination of the surface TiO_2_, which produces two types of carriers: an electron (e^−^) and a hole (h^+^), followed by oxidation of a donor molecule adsorbed on TiO_2_ by the photo-induced hole. The strong oxidation power of the holes enables the production of hydroxyl radicals (·OH) via the oxidation of water. Then, the adsorbed oxygen can be reduced by the promoted electron to form a superoxide ion (·O_2_^−^). NOx oxidation is a complex process involving several steps and intermediate species including the reduction of previously oxidized species. The NOx oxidation mechanism is summarized in Table 1 and Figure 1 (where hv: ultraviolet radiation, Site^(**): surface of TiO_2_ and OH: hydroxyl radical). Extensive studies have been conducted to analyze the oxidation mechanism of TiO_2_ under various experimental conditions [16,17,18,19,20,21].

In the past few decades, TiO_2_ has been used as a building material for atmospheric NOx removal by exploiting the NOx oxidation mechanism of TiO_2_ [22,23,24,25]. Chen and Poon [26] provided an overview of the development of photocatalytic construction materials in terms of academic achievements and practical applications. They determined that photocatalytic substances can be successfully applied to commercialized construction materials such as concrete, glass, paints and various types of cementitious materials. Seo and Yun [27] studied the nitric oxide (NO) removal performance of TiO_2_ cementitious materials under wet conditions. They found that the NO removal and absorption rates under dry conditions is higher as TiO_2_ particles come smaller. The recovery rate changed during evaporation under wet conditions, and three distinct phases (rapid recovery, stationary reaction and final recovery) were observed. In addition, they determined that the rate of NO removal was reduced when the photocatalytic cementitious material was rewetted. Song et al. [28] studied the NOx removal performance of TiO_2_-based coating materials used as paint. They demonstrated that sufficient NOx removal can be achieved although the NO gas concentration and UV-A irradiance are smaller than the experimental conditions of the ISO 22197-1: 2007 standard when their TiO_2_ photocatalyst-infused coating is used. Further, they determined that sufficient NOx removal can be achieved under adverse climatic conditions such as winter and gloomy days using TiO_2_ photocatalyst-infused coating materials. Based on the ISO 22197-1: 2007 standard, empirical studies on NOx removal efficiency and durability settings were conducted comprehensively in an indoor laboratory.

Recently, the extensive studies on the NOx removal efficiency and durability of TiO_2_-based construction materials have resulted in the development of methods for deriving quantitative results in outdoor environments for more practical purposes [29,30,31,32,33,34]. Guerrini [35] analyzed the NOx removal performance of photocatalytic cement-based paint practically by demonstrating the findings of two environmental monitoring campaigns involving the study of NOx levels measured before and after the reconstruction of Rome’s “Umberto I” tunnel. The results obtained by Guerrini indicate that the photocatalytic treatment of the Umberto I tunnel vault with cementitious paint resulted in effective pollution abatement, as shown by the lower concentrations found after the renovation. Yu et al. [36] investigated the performance of a mineral-based transparent air-purifying paint. The efficiency of this photocatalytic paint for removing air pollutants was determined in a laboratory setting using the ISO 22197-1 procedure. Subsequently, outdoor monitoring over a 20-month period was conducted to determine the air pollution removal efficiency under realistic conditions. They proposed a new protocol using the measured nitrate nitrogen (NO_3_^−^) produced by the photocatalytic oxidation of NO to monitor the air pollutant removal efficiency. Cordero et al. [37] evaluated the NOx removal efficiency of 10 selected materials using two pilot-scale demonstration platforms installed at two separate locations. The materials were exposed outdoors for measuring ground-level NO and nitrogen dioxide (NO_2_) concentrations over a one-year period. The pollutant removal efficiency of the materials was determined by comparing them to simultaneously measured concentrations of reference, nonactive materials.

Even though current studies on practical photocatalytic materials on a real scale have proposed their own distinctive methodology, there remain several challenges to providing an accurate assessment of NOx removal efficiency. The modified methodology based on the ISO 22197-1 standard addresses the following challenges. It is necessary to prove that there is a similar trend of quantitative NOx outcomes derived from indoor and outdoor experiments to confirm the NOx removal performance of photocatalytic materials in an outdoor environment. Further, the size of a real construction material, which is a trend indicating an increase in the amount of NOx observed as the size changes, should be demonstrated to certify the NOx removal efficiency of photocatalytic materials.

To this end, the NOx removal performance of photocatalytic construction materials is demonstrated in this study using two different types of experiments in both indoor and outdoor environments.

(1)A photoreactor test was conducted to assess the NO removal performance of construction materials such as coatings, paints and shotcrete using a modified ISO 22197-1 method. The UV irradiation of the outdoor environment was analyzed, and the experiment was conducted in an indoor laboratory under UV irradiation identical to that of the outdoor condition. Subsequently, photoreactor tests were conducted on construction materials applied to actual buildings located in Seoul, South Korea. The NO removal performance at the real scale was demonstrated by confirming that the trends of the indoor and outdoor environments showed comparable results.(2)In this experiment, the NOx removal performance was analyzed by assessing the amount of NOx ions remaining in the water after washing the surface of the specimen artificially exposed to NOx in the laboratory. The preliminary test used two same-sized specimens according to the specification in the ISO 22197-1 standard; the specimens were enlarged to the size of building materials. The experimental conditions were confirmed when the NOx removal performance was found to increase similarly to the tendency of the specimen to increase in size. The enlarged specimen was used for a field experiment by applying a modified method from the ISO 22197-1 standard.

## 2. Preliminary Test for Measuring NOx Removal

### 2.1. Experimental Setup

Various types of photocatalytic materials were used to analyze NOx removal performance (e.g., paint, coating material and shotcrete). Each photocatalytic material was prepared by mixing anatase-based TiO_2_ powder (NP-400 (Product Overview of Photoctalytic Powder NP-100. http://www.btfgreen.com/bbs/board.php?bo_table=product1&wr_id=4 (accessed 23 May 2021))from Bentech Frontier, Jeollanam-do, South Korea) with a commercial material. The mixing ratio of anatase-based TiO_2_ powder for each photocatalytic material is summarized in Table 2. In the table, composition of each photocatalytic material is specified in order to enhance the understanding of the results of this study. TiO_2_ powder was added to each commercial material. A fluid ceramic binder was used in advance to prevent substances in the paint from interfering with the photocatalytic reaction by chemically affecting the TiO_2_ powder. Other components of the paints are similar to those of common paint. The optimal photocatalytic reaction was elicited by adjusting the mixing ratio. Synthetic resin emulsions are used as a binder to increase the photocatalytic performance of shotcrete.

### 2.2. Photoreactor Test

We performed a preliminary test in the laboratory to measure the amount of NO removal from various types of construction materials such as coatings, paints and shotcrete using the photocatalyst before conducting the field application experiment. A preliminary test was conducted following ISO 22197-1 to verify the NO removal performance of the photocatalytic product specimens quantitatively. The NO injections ranged from 37.8–38.6 micromole (μmol) on a 5 h basis and 8.61 μmol on a 1 h basis in shotcrete measurement. A test condition of high NO gas concentration was prepared by connecting a hose line for gas injection and a bag containing NO gas while the flow rate of 1 ppm of NO gas was set to 1 L/min. Temperature was set to be 25 ± 2.5 °C and relative humidity was set to be 50% according to the ISO 22197-1 standard.

Figure 2 shows the apparatus used for NO removal analysis (CM2041, Casella, London, UK). In the ISO standard, the requirements of the analysis apparatus are clearly stated, and the apparatus used in this study is suitable for the ISO standard. For this test, UV-A with a wavelength range of 300 nm to 400 nm was used as the light source. This study followed the ISO standard’s content to use black light lamps with wavelength ranges of up to 351 nm. For the test, the ultraviolet light irradiance (UV irradiation) was kept constant at 10 W/m^2^. The NO removal performance of the photocatalytic coating and photocatalytic paint was measured for 5 h each, and that of photocatalytic shotcrete for 1 h each.

Table 3 lists the NO removal results for each photocatalytic construction material. Even with a 1-h measurement, the performance level of NO removal from photocatalytic shotcrete was higher than that of other materials.

Even though the ISO 22197-1 standard officially recommends setting UV irradiation as 10 W/m^2^, this study performed an additional test to determine UV irradiation identical to outdoor UV conditions for a more empirical experiment. We intended to determine the level of actual UV light incident on a vertical wall where the photocatalyst is applied because the UV light for photocatalytic action varies in the field whereas it is constant in the laboratory. As the photocatalyst is applied to the wall to which the reactor is attached, UV light on the southern wall, eastern wall and floor is measured on clear winter days. In the preliminary test, the NO removal amount on the eastern wall was measured in the field to determine the amount of NO injection (see Figure 3). A self-made photoreactor with a length of 300 mm and a width of 500 mm was used for the field measurements; this photoreactor is three times larger than the reactor prescribed in the ISO-22197-1 test method. The transparent window material was composed of quartz to facilitate the passage of ultraviolet light. The photoreactor was installed on the rooftop of the building with an NO injection of 3 ppm. Further, the NO removal amount was measured in the preliminary test to determine the appropriate level of NO injection for use in the field test.

Figure 4 shows the UV irradiation analysis results performed on the southern, eastern and floor surfaces between 9 a.m. and 4 p.m. at on 21 March. On the southern wall, the UV irradiation reached the highest level at noon and decreased gradually. On the eastern wall, the UV irradiation was relatively lower than that on the southern wall and floor surface, which indicates that the highest irradiation level was 6 W/m^2^. The UV irradiation is the highest on the floor because the construction materials receive sunlight when they are attached horizontally towards the sky. Thus, the installation strategy for photocatalytic materials needs to consider solar insolation efficiency.

### 2.3. Washing Water Experiment

This experiment involves the quantitative analysis of NO_2_ and NO_3_^-^ ions contained in washing water after washing the surface coated with photocatalytic paint with distilled water after the specimen is exposed to UV light for a certain period.

This quantitative experiment is conducted to determine the amount of NOx removed by the actual photocatalytic reaction. The experiment based on the washing water recovery method specified in Item 8.3 of ISO 22197-1 was conducted (Figure 5). Distilled water (50 mL) was used, and the washing time was set to 1 h. Elution operations were performed twice, and ions were measured via chromatography (Figure 6). Following ion measurement, the removed amount of NOx was calculated based on the number of ions eluted by the formulas specified in Item 9.7 of the ISO 22197-1 test method.
(1)$$n=n_{w1}+n_{w2}$$
(2)$$n_{w1}+n_{w2}=V_{w1}\left(p_{NO_{3}^{-},w1}÷62+p_{NO_{2}^{-},w1}÷46\right)+V_{w2}\left(p_{NO_{3}^{-},w2}÷62+p_{NO_{2}^{-},w2}÷46\right)$$
where *n*, *v*, $p_{NO_{3}^{-}}$ and $p_{NO_{2}^{-}}$ denote the number of moles of nitric acid eluted from the specimen, amount of distilled water recovered for washing, concentration of nitric acid eluted from the specimen and concentration of nitrous acid eluted from the specimen, respectively.

In this study, several experimental conditions were reviewed to modify the existing test methods of ISO 22197-1 for outdoor field test conditions. First, the amount of water washing was analyzed. If more washing water than necessary is used, it can dilute the ion concentration and make the measurement difficult; separate enrichment work is required to solve this problem. Second, a long washing time can increase the time required to process the specimen, which can result in an excessively long experimental schedule. The ISO-based specimens were washed following the test method specified in ISO 22197-1.

Before the field test, a few experimental variables were modified to examine that-the impact of the simplification of ISO 22197-1 method may have a significant effect on deriving the experimental results. Thus, the amount of washing water was adjusted to 20 mL for each washing considering the quantity of samples to be measured via ion chromatography; the washing time was limited to 5 min. The specific experimental conditions are addressed in Table 4.

The outdoor preliminary tests were conducted to check whether the test results showed a similar tendency to that observed in the laboratory experiments. A wall with photocatalytic paint was presumed, and common aqueous paint and photocatalytic paint were applied to ordinary concrete specimens. These specimens were then ex-posed to the same outdoor conditions and washed under the same conditions as the ones used in the preceding experiments to check the state of ion detection. For the pre-liminary outdoor experiments, specimens were installed on the roof of an apartment in Seoul, Korea. The specimens were exposed to natural conditions for a certain period, collected and washed by dipping in 20 mL of washing water for 5 min. The washing water was then collected to analyze the ion number using ion chromatography The size of the specimen is 49.5 mm in width and 99.0 mm in length, and the thickness of the specimen is 5 mm. The specimen is made by cutting the large plate with photocatalytic paint to the size specified above. The Specimen A and B were made by applying photocatalytic paint (containing about 3.5% of TiO_2_ component) to the surface of a plate made of concrete. In this study, the following pre-process is conducted on the specimens. First, in order to remove organic matter remaining on the surface, UV rays are irradiated for 16 h. The irradiated UV rays are10 W/m^2^ which can sufficiently decompose organic matter on the surface.

Two photocatalytic specimens, A and B, were used to conduct the ISO 22197-1 experiment to verify the activity for the preliminary test. Figure 7 shows the test results for the two types of specimens. In order to sufficiently supply NO gas in the reactor and react sufficiently with UV-A, lag occurred at the beginning. Since the experiment time is 5 h, it shows constant removal performance up to 300 min and shows the following results after 300 min. Specimen A showed an active photocatalytic reaction, with an NO removal rate of approximately 12%; specimen B showed a removal rate of approximately 5%, which is approximately half of that of specimen A.

Table 5 shows the results of analyzing NO removal of the NO attached to the surface during the test performed using the ISO 22197-1 measurement method and the nitric acid ionic weight detected in the washing water. For this experiment, the amount of washing water was 20 mL, and specimens were eluted five times for 5 min.

NO removal results and recovered nitric acid ionic weights from each result showed different trends. However, the recovery rate was found to be less than 50%, which differs from the calculated NO removal amount. Based on this measurement method, it can be concluded that the conditions for washing the specimen presented above are not correlated with the amount of NOx attached to the surface by photocatalytic activity. Therefore, only performing the test method specified in ISO 22197-1 can produce an overall error. In the above experiment, we used cured concrete specimens as the base materials to be painted; however, it is difficult to use them for panels exposed to outdoor environment. Even in the case of ISO specimens, it is expected that using mass-produced base materials for various experiments will yield more objective results. To sum up, the amount of washing water was adjusted from 20 mL to 50 mL for each washing considering the quantity of samples to be measured via ion chromatography; the washing time was also re-adjusted from 5 min to 1 h.

Considering the experimental conditions above, a cellulose fiber reinforced cement (CRC) board (from Byuksan corporation (Product Overview of CRC board used in this study. http://www.byucksan.com/01_product/product.asp?cate=001004002, accessed on 22 September 2021) Korea) was selected as the base material that satisfies the above conditions. The CRC board was fabricated by mixing natural pulp such as cellulose fiber, Portland cement and silica sand with water. The CRC board is a non-flammable (flame-retardant grade 1) construction material produced through an autoclave curing process after pressurizing to 10,000 tons. It is an eco-friendly asbestos-free construction material that exhibits only slight changes in length due to temperature variation, water resistance and noncombustibility (flame retardant grade 1). The experiment was conducted according to the method specified in ISO 22197-1 because washing the specimen under the conditions set previously does fully reflect the NO amount on the surface. The results are summarized in Table 6. The CRC-A and CRC-B were made by applying photocatalytic paint (containing about 3.5% of TiO_2_ component) to the surface of CRC board.

CRC boards are supplied through various processes such as manufacturing and storage, and therefore, it is difficult to verify if a significant amount of nitrogen oxides is already present. Therefore, removing nitrogen oxides contained in the base material is deemed necessary to use the CRC board as the base material. For the preliminary experiment, the experimental method with the modified specimens was conducted based on ISO standard methods established for outdoor field experiments. The experimental conditions and methods were as follows:Preparation of specimen
The base material of the specimen was unified as the CRC board.The CRC board used as the base material was washed sufficiently to remove pre-exiting nitrogen oxides.The test specimens are prepared with CRC boards; the photocatalytic paint was applied to these boards, and the general aqueous paint specimens were prepared as a control group.
Methods of analyzing NO removal and washing water.
NO removal was conducted in accordance with the ISO 22197-1 method.Concentrations other than those specified in ISO 22197-1 were not significant in the experiment; hence, the experiments were conducted only at the prescribed concentration of 1 ppm.The washing of the specimens that have undergone the ISO 22197-1 experiment are in accordance with the washing method specified in ISO 22197-1.The washing of ISO specimens exposed to the outdoor environment should be performed according to the washing method specified in ISO 22197-1.



## 3. Methodology for the Field Test

### 3.1. Photoreactor Field Test Method

In this study, photocatalytic materials applied to a real building were used to conduct field tests (Figure 8). After applying primers all over the southern wall of Building A, the photocatalytic coating agent was applied using rollers; the coating area was 297 m^2^. In Building B, we applied the photocatalytic paint by brushing and spraying onto the eastern wall, and the area of application was 889 m^2^. In Building C, the photocatalytic shotcrete was applied to a site of 61 m^2^, up to the second floor on the east side of the building. In the same manner as the preliminary test method, a reactor was installed in the area with the photocatalytic material applied to each building to measure the NO amount of injection and removal for more than 5 h. We simultaneously measured the UV irradiation. In the case of outdoor experiments, since it is impossible to adjust experimental variables such as illuminance, temperature and humidity, the experimental variables were carried out under natural experimental conditions.

Figure 9 shows installed photoreactor and UV radiometer used in the field experiments. The size of the UV radiometer is 50 × 300 mm, which is three times larger than the specimen defined in the ISO method. Considering that the gas could evenly reach the surface of specimen, the space between the transparent plate and the inner specimen was set to 5 mm. The photoreactor was set to 3 ppm as the specimen size was increased by a factor of three. We conducted a comparative evaluation of NO removal using three photocatalytic products based on the minimum criteria for photocatalytic product certification set by the Photocatalysis Industry Association of Japan.

### 3.2. ISO-Based Washing Water Field Test Method for Real-Scale Construction Materials

As the ISO-based specimen size is smaller than the size of the actual building material, a modified experimental method is used to perform practical and empirical analyses. To this end, the established methods are summarized as follows.

It is difficult to fabricate panels made for outdoor exposure using concrete. Hence, the CRC board reviewed in the ISO specimen tests was used as the base material. The size of the CRC board was set to 800 × 900 mm. The panel simulates the photocatalyst-coated surface for the convenience of the experiment because collecting washing water from the wall where photocatalytic paint is applied is difficult; this facilitates the identification of the amount of NOx removed from the surface by exposing it to the outdoor environment. Therefore, considering the field situation, we applied a washing method before measuring instead of dipping the specimens as prescribed in the ISO method. Figure 10 shows that the surface was evenly washed using a sprayer containing 2 L of distilled water. The quantity of distilled water is determined based on the empirical experiment, which indicates that 2 L of distilled water can fully wash the ions attached on the surface of CRC board.

The experiment was conducted from 25 October to 6 December. The reason why the outdoor experiment of this study was conducted during above period is to prevent the experimental results from being biased due to weather conditions such as excessive rainfall or insufficient rainfall. This experiment was conducted in Seoul, the capital of Korea. Seoul’s meteorological characteristics are that the rainfall is higher than other periods due to the influence of rainy seasons and typhoons in summer season (June to September). In winter season (December to February), snow falls and the temperature is low, making the walls coated with photocatalytic paint to freeze. These seasonal characteristics are expected to have an impact on the experimental results of this experiment. Therefore, in order to prevent bias in the experimental results due to the previous seasonal characteristics, the experimental period was set from 25 October to 6 December.

We performed washing up to five times as in the preliminary test to check the concentration of ions contained in the collected washing water; the results showed that the concentration was significantly lowered as the number of washings increased and that the concentration after the second washing was not significant. Therefore, the panels were only washed twice in the actual experiment. The experiments were conducted on the rooftop of a building adjacent to the roadway because measurements immediately next to the roadway were not feasible. The following field experimental conditions determined through preliminary experiments were used.

Panels with photocatalytic paint applied
The material of the panel was CRC board.The paint was applied to the panel after washing and drying the CRC board.The control group was prepared with a CRC board with general aqueous paint applied.
Methods of washing and analyzing the panels
A total of 2 L of washing water was used for each panel.The number of washes was limited to two; the washing intervals were determined by considering the weather conditions.The panel was washed by spraying.A separate holder and recovery bin was designed to collect rainwater.The number of ions in the washing water were measured via ion chromatography.


## 4. Results of NO Removal Analysis Experiment

### 4.1. Photoreactor Test Results

The results for the three experimental construction materials in each field are listed in Table 7. The NO removal amount on a clear day was more than twice that on the cloudy day for all three buildings. Assuming that the NO removal was converted to the ISO test reactor size of 50 cm², Building C with photocatalytic shotcrete applied had the lowest NO removal amount of 0.87 μmol/50 cm². However, it is still higher than that specified by the Photocatalysis Industry Association of Japan performance standard (0.5 μmol). Building C received less UV irradiation compared to buildings A and B because of the shade formed for a certain period of time. The preliminary test showed the highest NO removal performance of the photocatalytic shotcrete; however, the field measurement showed the lowest performance. As the NO removal performance is highly influenced by weather conditions, conditions such as the bearing of the building and its relationship with adjacent buildings need to be considered when applying photocatalytic products as exterior construction materials. Compared to preliminary test results measured in the laboratory, the field test results of Buildings A and B showed higher NO removal performance. For Building C, lower NO removal occurred in the field test than in the laboratory, which can be attributed to the difference in UV irradiation, which was 10 W/m^2^ in the preliminary test and less than 10 W/m^2^ in the field measurement.

### 4.2. Washing Water Field Test Results of Specimens

A specimen experiment was conducted according to the ISO 22197-1 method to determine how the amount of NOx removed by the photocatalyst is reflected in the washing water. The experiment was conducted with the specimen applied with aqueous paint as a comparison group and four types of specimens applied with photocatalytic paint. The specimens were classified into aqueous and photocatalytic paint specimens to facilitate the distinction of experimental specimens.

The size of the specimen here is 800 mm in width and 900 mm in length, and the thickness of the specimen is 5 mm. Photocatalytic paint containing the identical amount of TiO_2_ powder (about 3.5%) was applied to the surface of CRC boards to create Specimen A, B, C and D. For the photocatalytic paint specimens, newly prepared specimens A and B for this experiment and specimens C and D with photocatalytic paint applied approximately a month ago were used. These CRC specimens also went through the pre-process to remove organic matter remaining on the surface of CRC boards, UV rays are irradiated for 16 h. The irradiated UV rays are10 W/m^2^ which can sufficiently decompose organic matter on the surface. Each specimen was washed four times; however, only the ions in the first and second washing water were added and analyzed as prescribed in ISO 22197-1. Table 8 summarizes the results of the ISO 22197-1 experiment for the five specimens used for NOx removal with the results of washing twice.

Experimental results with the aqueous paint showed that the NOx removal amount was 0.08 μmol, which resulted in a weak removal effect; the washing water test results showed a small NOx removal amount of 0.53 μmol. The results obtained using specimens A and B for this experiment show that the calculated NOx removal amount and the amount of eluted nitrogen oxides are similar; in the case of existing specimens C and D, the results show a slightly more eluted NOx than the removed NOx. Thus, the results indicate that the amount removed by the ISO evaluation method is similar to the amount of elution measured from the analysis of the washing water. However, for existing specimens previously exposed to the outdoor environment, the NOx contained—whether through adsorption or reaction—was not completely removed during the pretreatment process. This is believed to have a partial effect on the elution into the washing water. When the specimen was washed sufficiently before the experiment, the error shown in the existing specimen is expected to decrease. In addition, it can be inferred that nitrogen oxides were sufficiently removed through the washing water experiment.

The specimen was placed outside the building from 3 June 2019 to 12 June 2019; during this period, the specimen was moved indoors on rainy days to prevent the surface from being washed by rainwater. The comparison results of the values obtained from experiments and the specimens exposed under the above conditions according to the ISO test method are listed in Table 9.

The specimens were exposed outdoors for five days in the apartment after washing was completed for the ISO experiment. For the general aqueous paints, 3.88 μmol of NOx was removed, whereas photocatalytic paints showed significantly better NOx elution. Some elution was found in the case of general aqueous paints; however, this was attributed to the result from surface adsorption rather than photocatalytic reaction. Further, for the photocatalytic paint with photocatalytic activity, we detected a large amount of elution in a definite contrast. Thus, we confirmed that the ISO NOx removal amount was equal to the value of the nitrogen oxide elution and that the photocatalytic performance showed a significant difference when the photocatalytic specimens were exposed in the field.

### 4.3. Washing Water Field Test Results of Photocatalytic Panel

An ISO-based washing water field experiment was conducted for real-scale construction materials under the same conditions as the specimen experiments.

Figure 11 shows panels installed on the roof of an apartment. The installation period is the same as that for which the ISO specimens were exposed outdoors; further, the panels were moved and kept indoors on rainy days to prevent the surface from being washed by rainwater. All panels were washed together under the same conditions, and all washing water was collected thoroughly and labeled (see Figure 12).

Table 10 summarizes the results of measuring the ionic value of the washing water of panels exposed on the roof of apartments in Seoul, Korea. In this study, three different types of panels are used for field test. First one is panel coated with aqueous paint. S1 is the panel that is newly coated with photocatalytic paint before the experiment day. S2 and S3 photocatalytic panels are created before two weeks before the experiment day. After exposing the panel to the outside, the surface was washed with water and reused.

A clear difference is observed between general aqueous paint and photocatalytic paint in terms of washing water for the CRC panels installed in the apartment. NOx removed by photocatalytic effects remains in an ionic state on the surface of the photocatalyst paint, and they are eluted by water washing and released into ionic nitric acid and nitrous acid. The amount of NOx ions eluted by water washing, which was investigated based on the NOx removal performance in the ISO test method, was almost the same as the amount of NOx removed by photocatalytic action. These results indicate that the amount of NOx ions in the washing water can be analyzed to quantify the amount of NOx removed by the photocatalyst. The NOx ions were observed in the washing water of general aqueous paint; however, only a few were detected in the washing water of photocatalytic paints. When tested and exposed to a real outdoor environment, the results indicated that photocatalysts are effective for removing NOx in the real world.

The results of s1 and s2 have the following implications for the NOx reduction performance of the photocatalytic paint used in this study. First, it shows that contaminants adhering to the surface are sufficiently removed just by washing the surface of the wall coated with photocatalytic paint with rainwater. Photocatalytic paint can maintain its NOx removal performance even after a certain period of time has passed since it is actually installed on the exterior wall of a building. Through this experiment, the durability of the performance when the photocatalytic paint was applied to an actual building is demonstrated.

Table 11 shows the amount of NOx that can be removed per day; it is calculated from the panel test results above. The average daily removal of the panel was calculated by averaging the installation days, excluding the rainy days when there was almost no UV light irradiation. Further, the estimated annual removal of the photocatalytic panel was calculated by excluding the average annual rainfall days in Korea. Moreover, the annual removal amount corresponds to the amount of NOx that can be removed per photocatalytic panel unit area.

## 5. Conclusions

A modified ISO 22197-1 standard method to analyze the NOx removal performance of photocatalytic construction materials in both indoor and outdoor environments was presented. The first experiment involved a photoreactor test conducted to assess the NO removal performance of construction materials such as coatings, paints and shotcrete. In the preliminary analysis, the impact of UV irradiation according to the direction at the outdoors was analyzed. For sunny days, the NOx removal performance (3.12–4.76 μmol/150 cm^2^·5 h) was twice as much as the ISO 22197-1 standard specification (2.03 μmol/150 cm^2^·5 h) in the real world. Even on cloudy days, it is slightly lower than that of the ISO 22197-1 standard specification; the NOx removal performance ranging from 0.68–1.89 μmol/5 cm^2^·5 h is shown, which is 63% of the indoor experiment results.

The second experiment was a washing water experiment, wherein the NOx removal performance was analyzed by assessing the amount of NOx ions remaining in the water after washing the surface of the specimen artificially exposed to NOx in the laboratory. The preliminary test employed two specimens, one of which was of the same size as that defined in the ISO 22197-1 standard, and the other was expanded to a size comparable to that of the construction materials. The experimental conditions were validated when it was discovered that the NOx removal efficiency improved in tandem with the specimen’s propensity to grow. The expanded specimen was then used in a field experiment using an adapted approach from the ISO 22197-1 standard. The washing water test was performed after sufficient washing of the ISO test specimen and exposure to the outdoors for 5 days in natural light. The washing test results indicated that general aqueous paint showed a NOx removal of 3.88 μmol, whereas that for photocatalytic paint was higher at 14.13 μmol. The amount of NOx removed by the washing water test was estimated to be 3.19 g/m^2^. This was demonstrated by the results of outdoor exposure, which is almost similar to the ISO standard test. Thus, it is possible to prove the direct correlation between the photocatalytic activity according to the ISO standard and the NOx removal performance.

The results of this study are expected to encourage the application of photocatalytic construction materials to real buildings with obvious NOx removal effectiveness. Empirically demonstrating that the real NOx removal effect increases when the small-sized specimen stated in the ISO standard is expanded to a size close to that of the actual construction material is a significant contribution from both academic and practical standpoints. Future work will include more extensive analysis using various types of photocatalytic construction materials at various sites for a longer experiment time.

## Figures and Tables

**Figure 1 materials-14-05717-f001:**
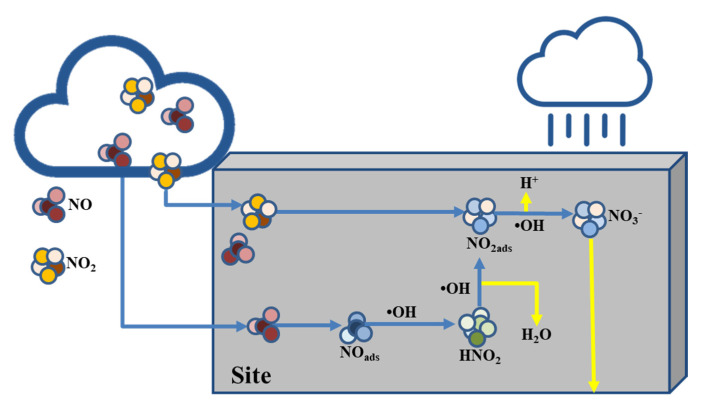
Mechanism for oxidation of NOx on the wall coated with TiO_2_.

**Figure 2 materials-14-05717-f002:**
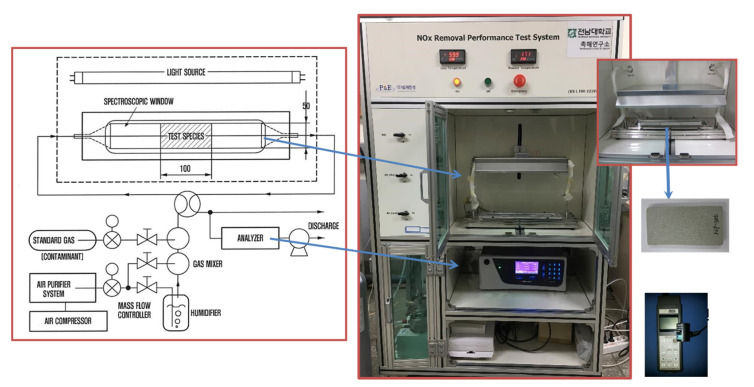
Apparatus used to measure NO removal as per ISO 22197-1 specifications.

**Figure 3 materials-14-05717-f003:**
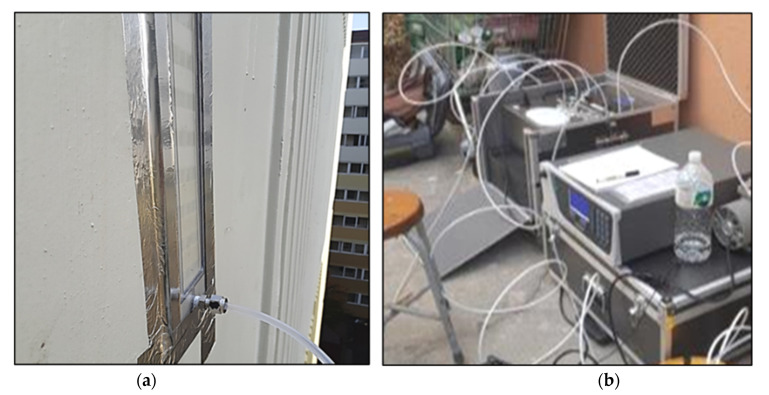
The experimental setup of photoreactor experiment on the wall. (**a**) Photoreactor with NO gas injection hoses on the eastern wall; (**b**) portable analysis device for NOx removal.

**Figure 4 materials-14-05717-f004:**
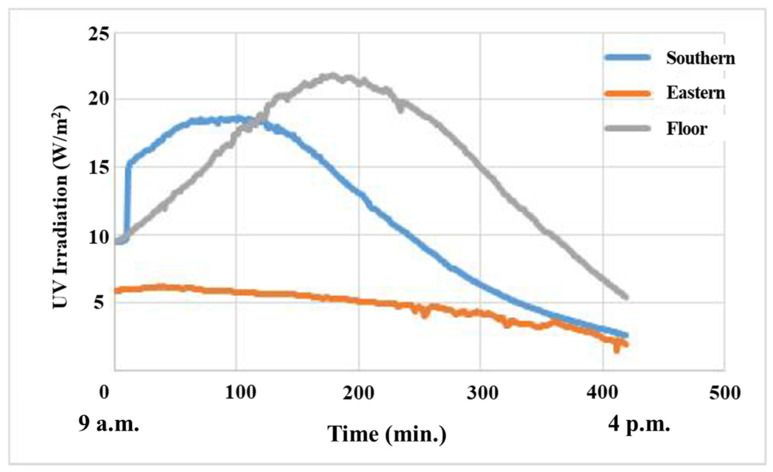
UV irradiation analysis results for southern, eastern and roof surfaces.

**Figure 5 materials-14-05717-f005:**
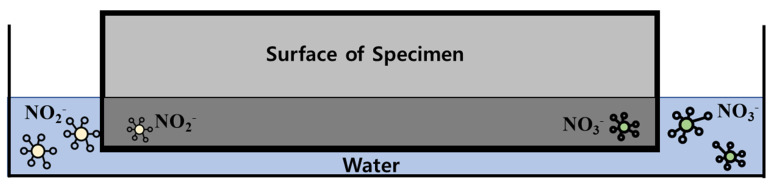
Overview of the washing water experiment of the specimen specified in ISO 22197-1.

**Figure 6 materials-14-05717-f006:**
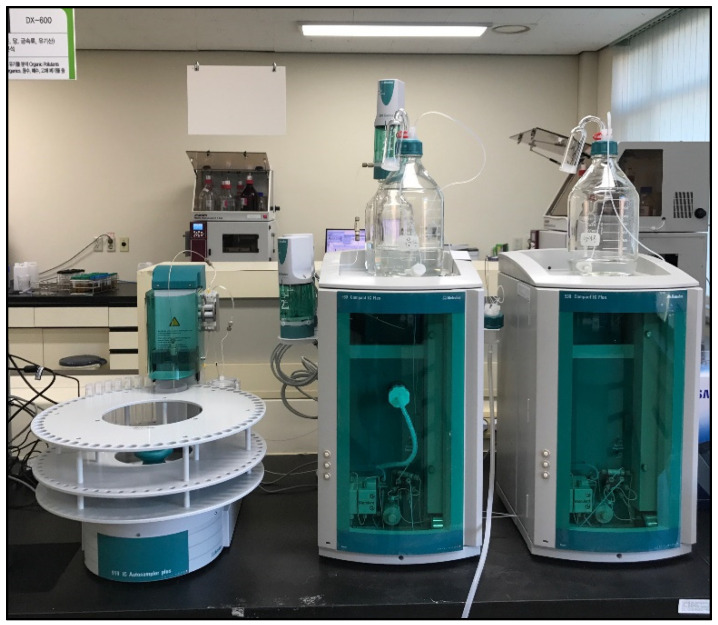
930 Compact IC Flex ion chromatography (940 Compact IC Flex, Metrohm, FL, USA).

**Figure 7 materials-14-05717-f007:**
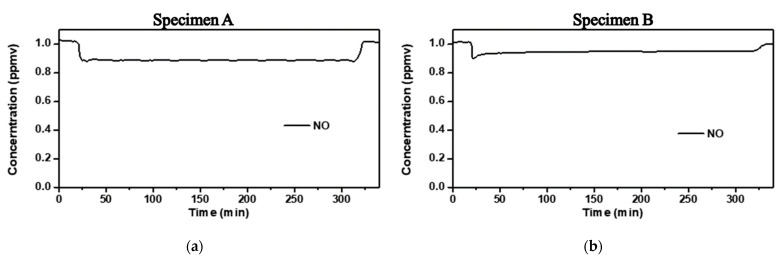
NO removal rate results for 6 h 30 min while UV-A is illuminated. (**a**) and (**b**), Ion chromatography result of specimen A and specimen B.

**Figure 8 materials-14-05717-f008:**
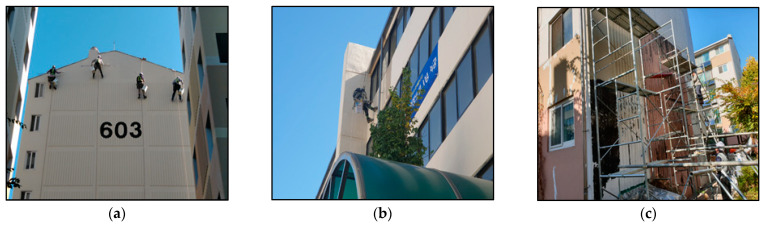
Photocatalytic materials applied on the real buildings in the field assessment. (**a**) Building A, photocatalyst coating; (**b**) Building B, photocatalyst paint; (**c**) Building C, photocatalyst shotcrete.

**Figure 9 materials-14-05717-f009:**
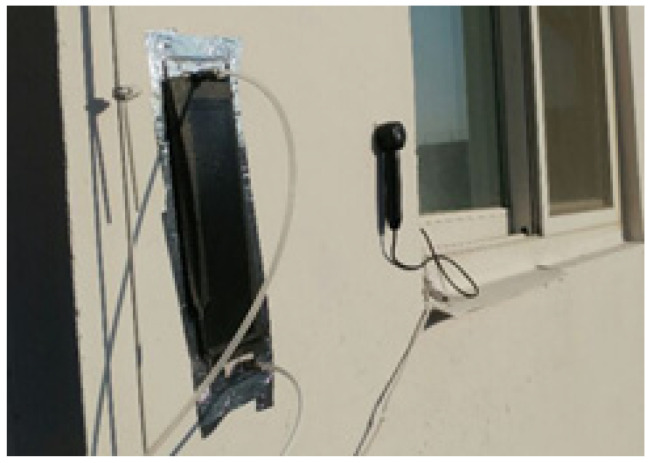
Installed photoreactor and UV radiometer on the rooftop.

**Figure 10 materials-14-05717-f010:**
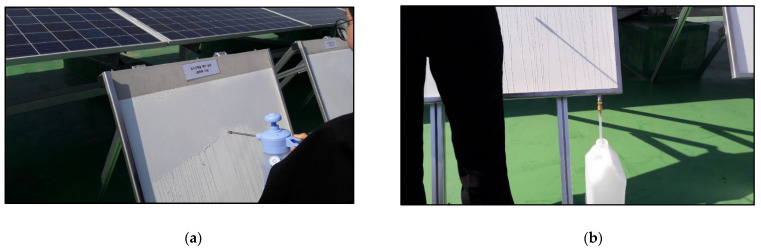
Spraying and collecting process of distilled water. (**a**) Spraying on the CRC panel; (**b**) collecting process of washing water.

**Figure 11 materials-14-05717-f011:**
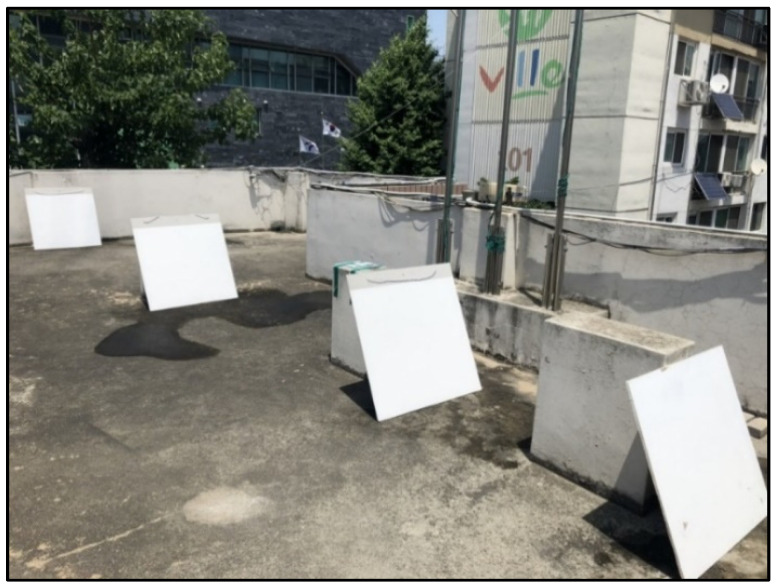
Specimens placed on the rooftop of the apartment.

**Figure 12 materials-14-05717-f012:**
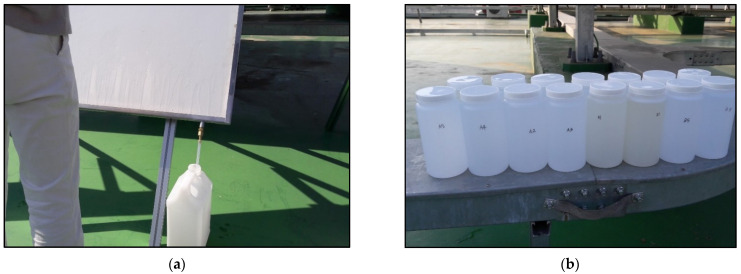
Collecting process and storage procedure of washing water. (**a**) collecting process of washing water; (**b**) collected washing water in seperate bottles.

**Table 1 materials-14-05717-t001:** Mechanism for oxidation of NOx on the surface TiO_2_.

Process	Formula
Activation	${\mathrm{TiO}}_{2}+\mathrm{hv}\;\to h^{+}+e^{-}$
	$H_{2}O\left(g\right)+{\mathrm{Site}\;}^{**}\to {\;H}_{2}O_{\mathrm{ads}}$
	$O_{2}\left(g\right)+{\mathrm{Site}\;}^{**}\to O_{2\mathrm{ads}}$
	$\mathrm{NO}\left(g\right)+{\mathrm{Site}}^{\;**}\to {\mathrm{NO}}_{\mathrm{ads}}$
Hole trapping	$H_{2}O+h^{+}\to \;\cdot \mathrm{OH}+H^{+}$
Electron trapping	$O_{2}\left(g\right)+e^{-}\to O_{2}^{-}$
Hydroxyl attack	${\mathrm{NO}}_{\mathrm{ads}}+2\cdot \mathrm{OH}\to {\mathrm{NO}}_{2\mathrm{ads}}+H_{2}{O\;\mathrm{NO}}_{\mathrm{ads}}+\cdot \mathrm{OH}\to {\mathrm{HNO}}_{3}$

Site **: surface of TiO_2_ and OH hydroxyl radical.

**Table 2 materials-14-05717-t002:** Composition of photocatalytic materials.

Material no.	Photocatalytic Coating		Photocatalytic Paint		Photocatalytic Shotcrete	
	Contained Chemicals	Proportion (%)	Contained Chemicals	Proportion (%)	Contained Chemicals	Proportion (%)
1	TiO_2_	1.75	TiO_2_	3.5	TiO_2_	3.5
2	Silicone compound	5.6	Fluid ceramic binder	7.2	Synthetic resins emulsions	2.2
3	Water	51.0	Others	89.3	Cements	17.2
4	Others	41.65	-	-	Others (Aggregates and water, etc.)	77.1

**Table 3 materials-14-05717-t003:** Photoreactor test results for NO removal rate at the preliminary test.

Materials	NO Injection (μmol)	NO Removal (μmol/50 cm^2^)	Measuring Time (Hour)	UV Irradiation (W/m^2^)
Photocatalytic coating	38.57	7.29	5	10
Photocatalytic paint	37.78	5.76	5	10
Photocatalytic shotcrete	8.61	11.28	1	10

**Table 4 materials-14-05717-t004:** The conditions of chromatographic ionic analysis.

Model	930 Compact IC Flex
Column	Metrosep A Supp 5, 250 × 4 mm
Eluent	3.2 mM Sodium carbonate, 1.0 M Sodium bicarbonate
Flow rate	0.7 mL/min
Inj. Volume	20 μL
Detection	Conductivity Detector

**Table 5 materials-14-05717-t005:** NO removal for each specimen, nitric acid ion weight and ratio analysis results.

Division	Removed NO (μmol)	Nitric Acid Ionic Weight (μmol)	Ratio (%)
Specimen A	4.74	0.96	48
Specimen B	2.03	0.34	41

**Table 6 materials-14-05717-t006:** NO removal results for CRC board panel experiments.

Division	NO Amount with Washing Once (μmol)	Accumulated Removed NO Amount (μmol)
CRC-A	0.81	3.57
CRC-B	1.11	2.84

**Table 7 materials-14-05717-t007:** Outdoor NO removal results for construction materials.

Materials		NO Injection Rate (µmol)	NO Removal (µmol/50 cm^2^)	NO Removal (µmol/50 cm^2^)	UV Irradiation (W/m^2^)	Measurement Date (Weather)
Building A: Photocatalytic coating		118.97	11.06	3.68	2–13	25 October 2018 (Sunny)
Building B: Photocatalytic paint	1st	107.82	10.03	3.34	6–14	14 November 2018 (Sunny)
	2nd	122.79	4.87	1.62	2–16	5 December 2018 (Cloudy)
Building C: Photocatalytic shotcrete	1st	95.84	6.78	2.26	1	13 November 2018 (Sunny)
	2nd	114.52	2.62	0.87	2–9	6 December 2018 (Cloudy)

**Table 8 materials-14-05717-t008:** Washing water field test results of five specimens.

Specimen Type	NOx Removal Amount (μmol)	Eluted NOx Amount (μmol)
Aqueous paint specimen	0.08	0.53
Photocatalytic paint specimen A	7.90	7.86
Photocatalytic paint specimen B	10.80	10.38
Photocatalytic paint specimen C	2.40	3.62
Photocatalytic paint specimen D	2.45	3.25

Refer to the preliminary test calculation formula for analyzing and calculating NO elution from ISO specimens.

**Table 9 materials-14-05717-t009:** Washing water field test results of specimens for which the surface was washed sufficiently.

Specimen Type	ISO NOx Removal Performance (µmol/50 cm²·5 h)	Elution Amount after ISO Experiment (µmol)	Elution Amount after Outdoor Exposure for Five Days (µmol)
Aqueous paint	0.08	0.53	3.88
Photocatalytic paint sample C	2.40	3.62	14.13

**Table 10 materials-14-05717-t010:** Washing water field test results of photocatalytic panels.

Photocatalytic Panels	Eluted NO_2_^−^, NO_3_^−^ Ions (μmol)
Aqueous paint	210
Photocatalytic panel (S1)	1469
Photocatalytic panel (S2)	475
Photocatalytic panel (S3)	504

**Table 11 materials-14-05717-t011:** Washing water field test results of photocatalytic panels.

Division	Daily Average NOx Removal Amount of Panel ^(1)^ (µmol)	Estimated Annual NOx Removal Amount of Panel ^(2)^ (µmol)	Estimated Annual NOx Removal Amount ^(3)^ (g/m^2^)
Aqueous paint	42.00	10,920	0.46
Photocatalytic panel (S1)	293.80	76,388	3.19
Photocatalytic panel (S2)	95.00	24,700	1.03
Photocatalytic panel (S3)	100.80	26,208	1.09

^(1)^ Total nitrogen oxides washed off from the panel/5 (number of days the panel was installed). ^(2)^ Panel daily average removal amount × 260 days (excluding the average number of rainfall days per year). ^(3)^ Estimated annual removal amount (μmol)/106 (calculated as mol) × 1.39 (area multiplier, m^2^) × 30 g/mol (nitrogen molecular weight).

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
