# Peer review of "Empirical NOx Removal Analysis of Photocatalytic Construction Materials at Real-Scale"

_materials, 2021, doi:10.3390/ma14195717_

Round 1

Reviewer 1 Report

The manuscript with a title of “Empirical NOx Removal Analysis of Photocatalytic Construction Materials at Real-Scale” researched NOx removal capabilities by several samples and even materials at real-scale. Such work was of some interests and organized very well. Therefore, I recommended accepting it after a minor revision. Some issues were listed as bellow:

  • When it came to photocatalytic NOx removal, humidity should be provided in each experiments;
  • Authors tested photocatalytic experiments from Oct 25 to Dec 6, why?
  • Authors compared NOx removal performance in lab and at real world. While, it was generally accepted the environments in lab and at real world were quite different and the experiment at real world was so complicated and affected by numerous aspects, such as the wind, some other contaminants, variation of humidity and light intensity. How did authors consider this question?

Author Response

Thank you for the review. I attached the response to your comment, so please refer to it in the review.

Reviewer 2 Report

This paper deals with NOx removal analysis of photocatalytic construction materials. The manuscript is quite well organized and report substantial new information on the subject. Only marginal aspects should be revised.

In particular, the authors should take into consideration the following points:

-Table 1 the authors should better explain/show the formation of the ?2? ????? compound.

-The authors should show the emission spectra of the used lamps or specify the working wavelengths.

-Figure 2: How the authors have performed this experiment? They should specify it in the experimental part.

Author Response

Thank you for the review. I attached the response to the comment, so please refer to it in the review.
